# Prevalence of Antibiotic-Resistant Seafood-Borne Pathogens in Retail Seafood Sold in Malaysia: A Systematic Review and Meta-Analysis

**DOI:** 10.3390/antibiotics12050829

**Published:** 2023-04-28

**Authors:** Omowale A. Odeyemi, Muhamad Amin, Fera R. Dewi, Nor Azman Kasan, Helen Onyeaka, Deyan Stratev, Olumide A. Odeyemi

**Affiliations:** 1Centre for Child & Adolescent Mental Health (CCAMH), University of Ibadan, Ibadan North, Nigeria; oaodeyemi@gmail.com; 2School of Nursing, Obafemi Awolowo University Teaching Hospital Complex, Ile Ife, Nigeria; 3Department of Aquaculture, Faculty of Fisheries and Marine, Universitas Airlangga, Jl. Mulyosari, Surabaya 60113, Indonesia; muhamad.amin@fpk.unair.ac.id; 4Research Centre for Applied Microbiology, National Research and Innovation Agency (BRIN), Cibinong 16911, Indonesia; era002@brin.go.id; 5Higher Institution Centre of Excellence (HiCoE), Institute of Tropical Aquaculture and Fisheries, Universiti Malaysia Terengganu, Kuala Terengganu 21030, Malaysia; norazman@umt.edu.my; 6HeTA Centre of Excellence for Food Safety, School of Chemical Engineering, University of Birmingham, Birmingham B15 2SQ, UK; h.onyeaka@bham.ac.uk; 7Department of Food Quality and Safety and Veterinary Legislation, Faculty of Veterinary Medicine, Trakia University, 6000 Stara Zagora, Bulgaria; deyan.stratev@trakia-uni.bg; 8Office of Research Services, Research Division, University of Tasmania, Hobart, TAS 7001, Australia

**Keywords:** seafood, antibiotic resistance, retail seafood, seafood-borne pathogens, microbial diversity

## Abstract

The objective of this study was to examine the frequency and extent of antibiotic-resistant pathogens in seafood sold in Malaysia, using a systematic review and meta-analysis approach to analyze primary research studies. Four bibliographic databases were systematically searched for primary studies on occurrence. Meta-analysis using a random-effect model was used to understand the prevalence of antibiotic-resistant bacteria in retail seafood sold in Malaysia. A total of 1938 primary studies were initially identified, among which 13 met the inclusion criteria. In the included primary studies, a total of 2281 seafoods were analyzed for the presence of antibiotic-resistant seafood-borne pathogens. It was observed that 51% (1168/2281) of the seafood was contaminated with pathogens. Overall, the prevalence of antibiotic-resistant seafood-borne pathogens in retail seafood was 55.7% (95% CI: 0.46–0.65). Antibiotic-resistant *Salmonella* species had an overall prevalence of 59.9% (95% CI: 0.32–0.82) in fish, *Vibrio* species had an overall prevalence of 67.2% (95% CI: 0.22–0.94) in cephalopods, and MRSA had an overall prevalence of 70.9% (95% CI: 0.36–0.92) in mollusks. It could be concluded that there is a high prevalence of antibiotic-resistant seafood-borne pathogens in the retail seafood sold in Malaysia, which could be of public health importance. Therefore, there is a need for proactive steps to be taken by all stakeholders to reduce the widespread transmission of antibiotic-resistant pathogens from seafood to humans.

## 1. Introduction

Fish, mollusks, crustaceans, and cephalopods have been reported to be healthy diets due to the high polyunsaturated fatty acids and essential trace elements [1,2]. As a result, many coastal countries have encouraged seafood farming to meet local and international needs.

Contaminated seafood could be a source of pathogens. Similarly, pathogen-harboring seafood could be rejected when exported. For example, Malaysia was one of the countries that had their seafood and seafood products seized by European countries due to the presence of antibiotics [3]. In a recent study, Malaysia was identified as one of the four countries (Malaysia, Thailand, Vietnam, and Indonesia) from Southeast Asia that had their seafood products rejected by 19 European countries due to pathogens from 1997 to 2020 [4]. Seafood-borne pathogens and antibiotic resistance [5] are considered global health problems as they threaten food security and global health [6]. In Europe, antibiotic-resistant pathogens have been reported to be responsible for more than 30,000 deaths yearly [7]. It was also predicted that by 2050, antibiotic-resistant pathogens might cause 10 million deaths per year globally and this could potentially cause a reduction of 2% to 5% (USD ~100 trillion) in gross domestic product [7]. In Malaysia, various studies have reported the incidence of seafood-borne pathogens. However, the pool prevalence estimate, the diversity of antibiotic-resistant seafood-borne pathogens, and the antibiotics that the pathogens are resistant to are not yet known. The use of systematic review and meta-analysis could help to pool the results of these studies.

Over the years, there has been an increase in the use of systematic review and meta-analysis to understand the overall prevalence of food-borne pathogens [8,9,10,11,12,13]. While a systematic review is used to synthesize evidence, meta-analysis is a quantitative statistical technique to summarize extracted data from primary studies. Both methods are combined to provide robust evidence that can help different stakeholders make informed decisions.

To the best of our knowledge, no systematic review or meta-analysis have been conducted to understand the overall prevalence of antibiotic-resistant seafood-borne pathogens in retail seafood sold in Malaysia. This study therefore aimed to synthesize evidence and summarize primary studies to describe the prevalence of antibiotic-resistant seafood-borne pathogens in retail seafood sold in Malaysia.

## 2. Results

### 2.1. Literature Search and Eligible Studies

Thirteen primary studies (Figure 1) met this study’s inclusion and exclusion criteria. In the primary studies, a total of 2281 retail seafoods were analyzed for the presence of antibiotic-resistant seafood-borne pathogens in retail seafood in Malaysia. As shown in Figure 2A–C, half of the studies (52%) were carried out in Selangor and 57% of the retail seafood was obtained from wet markets, while cephalopods were the least investigated retail seafood. It was observed that 51% (1168/2281) of the seafood was contaminated with pathogens. The results of the meta-analysis of the overall prevalence of pathogens are presented in Table 1.

### 2.2. Retail Samples Analyzed

Figure 3A–C shows the types of retail seafood that were investigated for the presence of antibiotic-resistant seafood-borne pathogens.

#### 2.2.1. Fish

A total of 781 retail fish including tilapia, catfish, mackerel, and herring were investigated by five studies (2, 4, 7, 8, and 9) for the presence of antibiotic-resistant seafood-borne pathogens. The distributions of the fish samples that were positive for pathogens are shown in Figure 4A.

#### 2.2.2. Mollusks

A total of 754 mollusk samples comprising cockles, mussels, crabs, and snails were analyzed by 7 primary studies (1, 3, 5, 10, 11, 12, and 13) for the presence of antibiotic-resistant seafood-borne pathogens. The distribution of the mollusk samples is shown in Figure 4B. Cockles/clams comprised 49% of the positive samples. Primary study 5 analyzed the highest (49.7%) number of samples, with 60% positive samples. The least analyzed mollusk was the mussel (1%).

#### 2.2.3. Crustaceans

A total of 711 crustacean samples comprising prawn and shrimp were analyzed by 4 primary studies (3, 6, 12, and 13) for the presence of antibiotic-resistant seafood-borne pathogens. The distribution of the crustaceans is shown in Figure 4C.

#### 2.2.4. Cephalopods

A total of 35 cephalopod samples comprising squid were analyzed by primary study 13 for the presence of antibiotic-resistant seafood-borne pathogens. The distribution of the cephalopod samples is shown in Figure 4D.

### 2.3. Pathogens

The overall prevalence of antibiotic-resistance seafood-borne pathogens in retail seafood in Malaysia is shown in Table 1.

#### 2.3.1. Fish

It was observed that 70% (550/781) of the fish samples were contaminated with *Aeromonas* sp., *Salmonella* sp., and *Vibrio* sp., while 303 confirmed isolates of *Salmonella* sp. (4%), *Aeromonas* sp. (19%), and *Vibrio* (77%) were reported in the 550 positive fish samples, as shown in Figure 5. A total of 57 *Aeromonas* (primary study 2) isolates comprising *Aeromonas hydrophila*, *Aaeromonas veronii* biovar sobria, and *Aeromonas caviae* were reported by primary study 2 (Figure 6A). *A. veronii* biovar sobria was more prevalent (84%) in the fish samples. For example, tilapia had 18/48 of *A. veronii* biovar sobria. The least represented *Aeromonas* species (*A. caviae*) was from the northern red snapper fish. A total of 11 confirmed isolates of *Salmonella* from primary study 4 (7 isolates), namely *Salmonella* serotype Agona, *Salmonella* serotype Albany, *Salmonella* serotype Corvallis, and *Salmonella* serotype Typhimurium, were reported (Figure 6B). It was observed that S. Corvallis was more prevalent (71%) in catfish (*Clarias gariepinus*) than in other fish samples. A total of 235 *Vibrio parahaemolyticus* from tilapia (*Oreochromis* spp.), catfish (*Clarias batrachus*), Indian mackerel (*Rastrelliger kanagurta*), and short mackerel (*Rastrelliger branchysoma*) were reported by primary studies 8 and 9 (Figure 6C).

#### 2.3.2. Mollusks

It was observed that 44.6% (336/754) of the mollusk samples were contaminated with *Vibrio* and MRSA (Figure 7).

A total of 384 confirmed isolates of *Vibrio* and MRSA were reported in the 336 positive mollusk samples. The number of pathogens was more than the positive samples because, in some of the studies, the different parts of the positive samples were analyzed. A total of 378 *Vibrio* isolates comprising *V. parahaemolyticus*, *Vibrio vulnificus,* and *Vibrio cholerae* were reported by 6 primary studies (primary studies 1, 3, 5, 11, 12, and 13). *V. parahaemolyticus* was isolated from cockles, crabs, and snails (Figure 8A). However, this pathogen was prevalent (88%) in cockles. The least common *Vibrio* species (*V. cholera*) was from the cockles. A total of 7 MRSA isolates from blood cockles/clams (*Anadara granosa*) and green mussels were reported by study 10 (Figure 8B).

#### 2.3.3. Crustaceans

It was observed that 37% (254/711) of the crustaceans were contaminated with *Vibrio* species. A total of 252 *Vibrio* isolates comprising *V. parahaemolyticus* and *V. cholerae* were confirmed and reported in primary study 2. *V. parahaemolyticus* was more prevalent (96%) in the crustacean samples (Figure 9) than *V. cholerae*.

#### 2.3.4. Cephalopods

It was observed that 80% (28/35) of the squids analyzed were contaminated by *V. parahaemolyticus* and a total of 28 confirmed isolates of the pathogen were reported.

### 2.4. Antibiotic Resistance

A total of 172 antibiotics were tested in all the studies. However, after deduplication, 46 unique antibiotics were identified (Table 2). The most tested antibiotics in the primary studies were Tetracycline and Chloramphenicol (92%), while 10 (77%) of the primary studies tested Ampicillin and Gentamicin, respectively. The results of the meta-analysis of the overall prevalence of antibiotic resistance based on the types of seafood and pathogens are presented in Table 3.

#### 2.4.1. Fish

Overall, *Aeromonas* species were resistant to 93% (14/15) of antibiotics tested (Appendix A). The *Aeromonas* species were resistant to Ampicillin (100%), Carbenicillin 96% (55/57), and Erythromycin 95% (54/57). The pathogen was least resistant to Chloramphenicol (5.26%). *Salmonella* species were resistant to 46% (6/13) of the antibiotics tested and were more resistant to Clindamycin 100% (11/11) and Rifampicin 100% (11/11), but least resistant to Spectinomycin 27%. *Vibrio* species were resistant to 86% (19/22) of the antibiotics tested. *Vibrio* species were more resistant to Ampicillin (85.11%; 200/235), Amikacin (51.49%; 87/235), and Penicillin G (38.72%; 28/235), but least resistant to Doxycycline and Levofloxacin (0.85%; 2/235).

#### 2.4.2. Mollusks

The *Vibrio* species were resistant to 94% (29/31) of antibiotics tested (Appendix A). The *Vibrio* species were more resistant to Ampicillin (88%) and least resistant to Amoxicilin-clavulanic acid (0.26%), while MRSA was resistant to 24% (4/17) of the antibiotics tested. MRSA was 100% resistant to Penicillin G and least resistant (28.57%) to Cefepime.

#### 2.4.3. Crustaceans

The *V. parahaemolyticus* from red prawn were resistant to 10 antibiotics, while isolates from banana prawn were resistant to all the 14 antibiotics tested. It was observed that 79% (77/97) of the isolates from red prawn and 84% (74/88) of the isolates from banana prawn were resistant to Ampicillin, while *Vibrio* species isolated from red prawn were least resistant (1.03%) to Imipenem and Gentamicin, and isolates from banana prawn were least resistant (3.41%) to Ampicillin-sulbactam, Gentamicin, and Imipenem (Appendix A).

#### 2.4.4. Cephalopods

The 28 *V. parahaemolyticus* isolates from squids were resistant to 54% (13/24) of the antibiotics tested. It was observed that 96% of the pathogens were resistant to Penicillin G, 79% resistant to Ampicillin and Cefazolin, while the least resistance (4%) was to Cefotaxime and Ceftazidime (Appendix A).

## 3. Discussion

A systematic review is used to synthesize the results of various primary studies, while the meta-analysis is used for the statistical analysis of the results derived from the primary studies [27]. Systematic reviews and meta-analysis are often combined to develop an unbiased assessment of available data [28]. Systematic reviews are important tools for decision-makers to summarize and synthetize the available evidence on a topic of interest [29]. In this study, we combined systematic review and meta-analysis on the occurrence and prevalence of antibiotic-resistant seafood-borne pathogens in retail seafood sold in Malaysia.

Research outputs are mostly published as journal articles that are housed in bibliographic databases. The most popular search databases are PubMed, Web of Science, Science Direct, and Scopus [30]. In this study, we used these databases and obtained thirteen studies which met our inclusion criteria. Half of the included primary studies were published by a public university—the University of Putra Malaysia in Selangor. It was not surprising that the University of Putra Malaysia published half of research outputs involving antibiotic-resistant seafood-borne pathogens in retail seafood sold in Malaysia. The Malaysian National Higher Education Strategic Plan Beyond 2020 structured all the twenty public universities into three categories, namely research universities, comprehensive universities, or focused universities [31]. The University of Putra Malaysia is one of the five research universities [32], and therefore, the university is expected to produce more research outputs. The retail seafood investigated mainly comprised fish, mollusks and crustaceans obtained from wet markets and supermarkets. Bivalve mollusks, shrimps, giant tiger prawns, and marine fish comprise the major seafood of aquaculture sector in Malaysia [33].

In a recent study, Malaysia was identified as one of the four countries (Malaysia, Thailand, Vietnam, and Indonesia) from Southeast Asia that had their seafood products rejected by 19 European countries due to the presence of pathogens from 1997 to 2020 [4]. Guardone el al. [3] also reported Malaysia as one of the countries that had their seafood and seafood products seized by European countries due to the presence of antibiotics. While various studies have been conducted in Malaysia examining the presence of antibiotic-resistant seafood-borne pathogens in seafood, there was a dearth of studies on the overall prevalence of the pathogens and the types of antibiotics the pathogens are resistant to Pigłowski [34] reported the presence of *Vibrio* in seafood imported from Malaysia from 2001 to 2004. Therefore, this study aimed to investigate the occurrence and prevalence of antibiotic-resistant seafood-borne pathogens in retail seafood sold in Malaysia, through a systematic review and meta-analysis of 13 primary studies from 1993 to 2021.

In the primary studies included in this current study, a total of 2281 seafood samples were analyzed for the occurrence of antibiotic-resistant seafood-borne pathogens. More than 50% of the samples were contaminated by antibiotic-resistant seafood-borne pathogens. The results of this study indicated the overall prevalence of antibiotic-resistant pathogens in retail seafood in Malaysia to be 55.7% (95% CI: 0.46–0.65). Our study showed that antibiotic-resistant *Salmonella* had an overall prevalence of 59.9% (95% CI: 0.32–0.82) in fish. Although *Salmonella* is not a typical aquatic organism, it is one of the most frequent pathogens causing gastroenteritis in people [35]. Production-related hygiene problems have been identified as the main cause of the prevalence of *Salmonella* in aquaculture environments and seafood products [36,37]. Novoslavskij et al. [38] found that *Salmonella* was prevalent in seafood products, especially fish, and the frequent serotypes recovered from seafood were *S. enterica* serovar Typhimurium, ser. Enteritidis, ser. Typhi, ser. Paratyphi B, and ser. Newport. *S*. Typhimurium and Enteritidis serotypes of *S. enterica* were predominant in human cases, while Paratyphi B and Typhi were as a result of contamination during handling of samples [38,39]. Ponce et al. [40] reported that *Salmonella* serovar Typhimurium is a frequent and increasing cause of human infection and is the predominant serovar in Malaysia, Thailand, and Vietnam. In a study by Hatha and Lakshmanaperumalsamy [41], 14 out of the 18 fish samples were *Salmonella*-positive. The highest incidence was seen in fish samples from the genera Mugilidae 21/86 (24.4%), Scopelidae 7/25 (28%), and Trachnidae 7/26 (26.9%). The authors suggested that *Salmonella* may grow more readily in certain types of fish due to high lipid content. Budiati et al. [37] studied the presence of pathogens in freshly caught and retail tilapia (*T. mossambica*) and catfish (*C. gariepinus*). The study showed that *S. enterica* serovars (Agona, Bovismorbificans, Corvallis, Typhimurium) and Albany, Agona, Corvallis, Stanley, and Typhimurium were highly prevalent, with 32/14 (43.8%) and 32/9 (28.1%). Several studies, including [39,42,43,44] and [45], have found *Salmonella* to be a pathogen with a high prevalence in fish. Both live and caught fish can be impacted by the presence of *Salmonella* due to the use of contaminated water, improper feeding techniques, contaminated feed, and unhygienic fish catch, handling, and shipping practices [38].

In cephalopods, *Vibrio* species were found to have an overall prevalence rate of 67.2% (95% CI: 0.22–0.94). *Vibrio* spp. can naturally be found in freshwater, estuarine, and marine habitats [10]. *Vibrio* spp. can cause cholera, septicemia, and gastroenteritis [46]. *V. parahaemolyticus*, *V. vulnificus*, and *V. mimicus* are examples of seafood-borne pathogens [47].

In mollusks, 384 confirmed isolates of *Vibrio* and MRSA were reported in the 336 positive samples, and six studies reported a total of 378 *Vibrio* isolates comprising *V. parahaemolyticus*, *V. vulnificus*, and *V. cholerae*. A recent study by Haifa-Haryani et al. [48] found 225 *Vibrio* spp. in cultured shrimp, with *V. parahaemolyticus* being the most prevalent species, isolated from 6 sampling locations. Seafood, particularly shellfish and bivalve mollusks, are common sources of *V. parahaemolyticus* worldwide. In our study, we found that 44.6% (336/754) of mollusk samples were contaminated with *Vibrio* and MRSA (Figure 7). This was similar to a study by Letchumanan et al. [49] that reported the prevalence of *V. parahaemolyticus* in retail seafood sold in Malaysia. Studies from other countries have also shown the presence of *V. parahaemolyticus* in fish, shrimp, and bivalve mollusks in Croatia (9.4%) [50], Sri Lanka (98%), Ecuador (81%), and Egypt (18%) [51,52,53]. Other studies have also reported that 24.2% of raw bivalve mollusk samples were positive for *V. parahaemolyticus* [2]. The prevalence of *Vibrio* in fish from Greece and Portugal was also reported [54,55]. Similarly, a study showed that most *V. parahaemolyticus* isolates in the Netherlands were isolated from clams, mussels, and oysters [2]. Similarly, a study conducted in other European countries found that *V. parahaemolyticus* primarily affected oysters (42.2%), mussels (33.1%), and clams (24.7%) [56]. This pathogen has been detected in the marine waters of several European nations, including Great Britain, France, Spain, Italy, and Slovenia [57,58]. Although *V. parahaemolyticus* is frequently associated with mollusks, it is also highly prevalent in fish, particularly in Asian countries [59]. The European Union currently does not have a microbiological standard for raw bivalve mollusks with regards to *V. parahaemolyticus*; however, it is recommended that mollusks taken from water contaminated with *Vibrio* sp. be tested for the presence of *V. parahaemolyticus* [54].

It is difficult to attribute antibiotic resistance in fish farms solely to human activity, as bidirectional flow of antimicrobial-resistant bacteria and genes can occur between the marine environment and human microbiota [60]. The *Aeromonas* species in this current study showed high resistance to 93% (14/15) of the antibiotics tested in the primary studies, with Ampicillin (100%), Carbenicillin (96%, 55/57), and Erythromycin (95%, 54/57) being the most resisted, while Chloramphenicol was the least resisted antibiotic (5.26%). Schar et al. [61] found that seafood-borne pathogens, including *Vibrio* and *Aeromonas* spp., had high rates of resistance to first-line antimicrobial classes and moderate-to-high rates of resistance to antimicrobial classes of last resort. *Aeromonas* spp. isolates from Western and Southern Asia have high levels of resistance to tetracycline, sulfonamide, aminoglycoside, monobactam, carbapenem, and cephalosporin antibiotics. This indicates that therapeutic options for controlling invasive or non-self-limiting *Aeromonas* infections may already be limited in this subregion compared to areas with lower resistance profiles [62,63]. The use of antibiotics in aquaculture practice could increase the resistance of seafood-borne pathogens to antibiotics [64].

## 4. Materials and Methods

This study was designed and carried out in six stages following a modified method of Preferred Reporting Items for Systematic Reviews and Meta-Analyses—PRIMA [11]. The stages are defining key terms, developing the research questions, identifying relevant studies, study selection, data extraction, and data analysis.

### 4.1. Stage 1: Definition of Terms

As used in this study, incidence is defined as the presence or occurrence of seafood-borne pathogens in retail seafood sold in Malaysia as reported in the primary studies [10]. Primary studies are the studies carried out by other researchers to collect the primary data that were used in this study. These primary studies have therefore already been published in referred journals. The primary studies involve collecting samples of retail seafood, transporting them to the laboratory, and analyzing them for the presence (incidence) and prevalence of antibiotic-resistant seafood-borne pathogens. Prevalence is the rate of occurrence of the pathogens in the samples analyzed from the sample population [10]. Population as used in this study was the various seafood samples investigated in the primary studies. In Malaysia, seafood can be purchased from the hypermarket, supermarket, or wet markets as live or dead aquatic animals. Hypermarkets are retailers of fast consumers goods and basic household needs such food, vegetables, and seafood within different sections [65]. This is similar to a supermarket with slight differences. In supermarkets, there could be the sale of items such as clothing and electrical products in addition to the sale of basic household needs such as food. Both hypermarkets and supermarkets operate every day. On the other hand, in a wet market, consumable goods are sold weekly, and this involves farmers bringing their produce, including seafood, directly from the farm for sale to the public. The food items sold are deemed to be fit for human consumption and they are usually cheaper in price compared to those that are sold in hyper- and supermarkets.

### 4.2. Stage 2: Developing Research Questions

The research questions guiding the study were based on the PCCO framework: P—population (retail seafood), C—concept (incidence of antibiotic-resistant seafood-borne pathogens), C—context (Malaysia), and O—outcome (prevalence). The following research questions were therefore formulated to guide this study:Are there seafood-borne pathogens in retail seafood sold in Malaysia?Are these pathogens resistant to antibiotics?What is the prevalence of antibiotic-resistant seafood-borne pathogens in retail seafood sold in Malaysia?

### 4.3. Stage 3: Identifying Relevant Primary Studies

To identify relevant primary studies to be used in this study, the following databases were searched: Web of Science, Science Direct, PubMed, and Scopus. The keywords used to search the databases for the relevant primary studies were “antibiotic resistance”, “fish”, “seafood”, “shellfish”, “crustaceans”, “finfish”, “antimicrobial”, and “antibiotic susceptibility”. These keywords were combined using Boolean operators “AND” and “OR” to ensure only primary studies relevant to this study were obtained. The primary studies were limited to Malaysia and primary studies published up to 2021 were considered.

### 4.4. Stage 4: Selection and Screening of Relevant Primary Studies

All the “relevant” studies obtained from the various databases were exported into Microsoft Excel and checked for duplicates. The titles and abstracts of the primary studies were screened based on the inclusion criteria below:primary study carried out in Malaysia;retail seafood samples were purchased from hyper-, super-, or wet markets;types of antibiotics used were stated;total number of samples (population) and positive samples were stated;method of isolation and identification of seafood-borne pathogens.

The primary studies excluded were based on the exclusion criteria below:primary studies involved diseased seafood;processed seafood products;studies involving only isolation and characterization;studies involving pathogens from culture environment fish farms or ponds;the studies were led by authors from other countries with Malaysian co-author(s).

### 4.5. Stage 5: Data Extraction

This stage of the study involved the descriptive summary extraction of the relevant primary studies that fulfill the aims of the systematic review and meta-analysis. The data extracted included author(s), journal, title, location of the study within Malaysia, year of publication, types and place of purchase of the seafood, and antibiotics investigated.

### 4.6. Stage 6: Data Analysis

The data analysis was carried out in two phases. In phase 1, a descriptive analysis was carried out Microsoft Excel (2022 version). The second phase involved obtaining the individual and pooled estimates of the prevalence of antibiotic-resistant seafood-borne pathogens in each retail seafood based on a random-effect meta-analysis model. However, the variation between the primary studies was evaluated using heterogeneity (I^2^). The presence of bias in the primary studies was determined, and the event rate was estimated at 95% confidence intervals using forest plots [10,11,66,67]. The analysis in phase 2 was carried out using Comprehensive Meta-Analysis (CMA) software [68]. The study protocol was registered with Open Science Framework Registries with the registration DOI https://doi.org/10.17605/OSF.IO/5A284 (created on 17 March 2023).

## 5. Conclusions

This study provided a comprehensive understanding of the prevalent antibiotic-resistant seafood-borne pathogens in retail seafood in Malaysia. There is a need for continuous monitoring of the use of antibiotics in aquaculture and in retail seafood. This will help to implement effective management strategies to minimize the spread of antibiotic-resistant pathogens. These strategies include improved hygiene practices, reducing the use of antibiotics, and stricter regulations on aquaculture and water management. Overall, these studies highlight the urgent need for a One Health approach that recognizes the interconnectedness of human, animal, and environmental health. By working together to address the issue of antibiotic resistance, we can help to ensure a sustainable future for both human and animal populations.

## Figures and Tables

**Figure 1 antibiotics-12-00829-f001:**
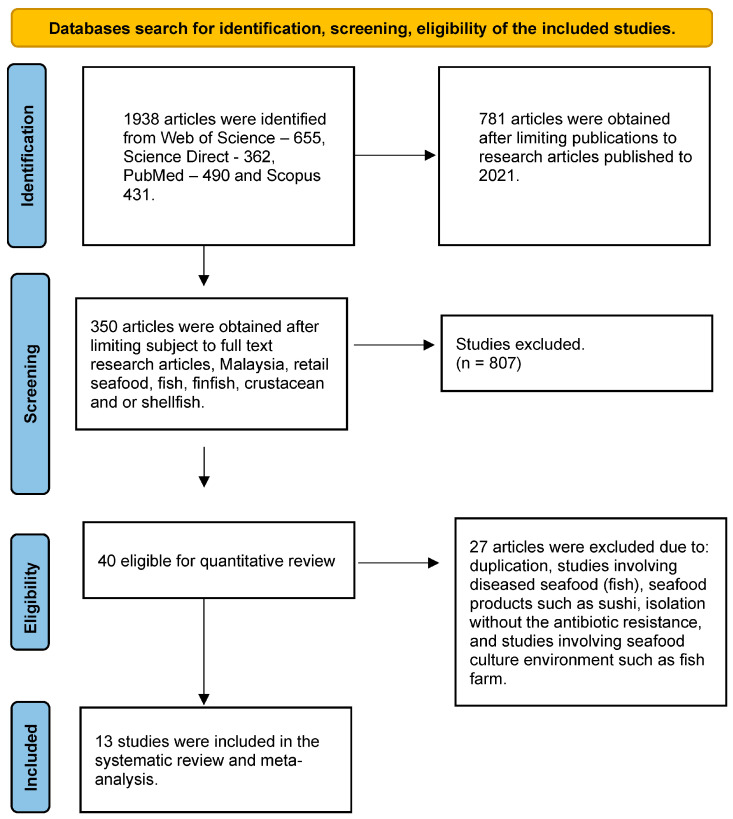
Flow diagram of selected studies.

**Figure 2 antibiotics-12-00829-f002:**
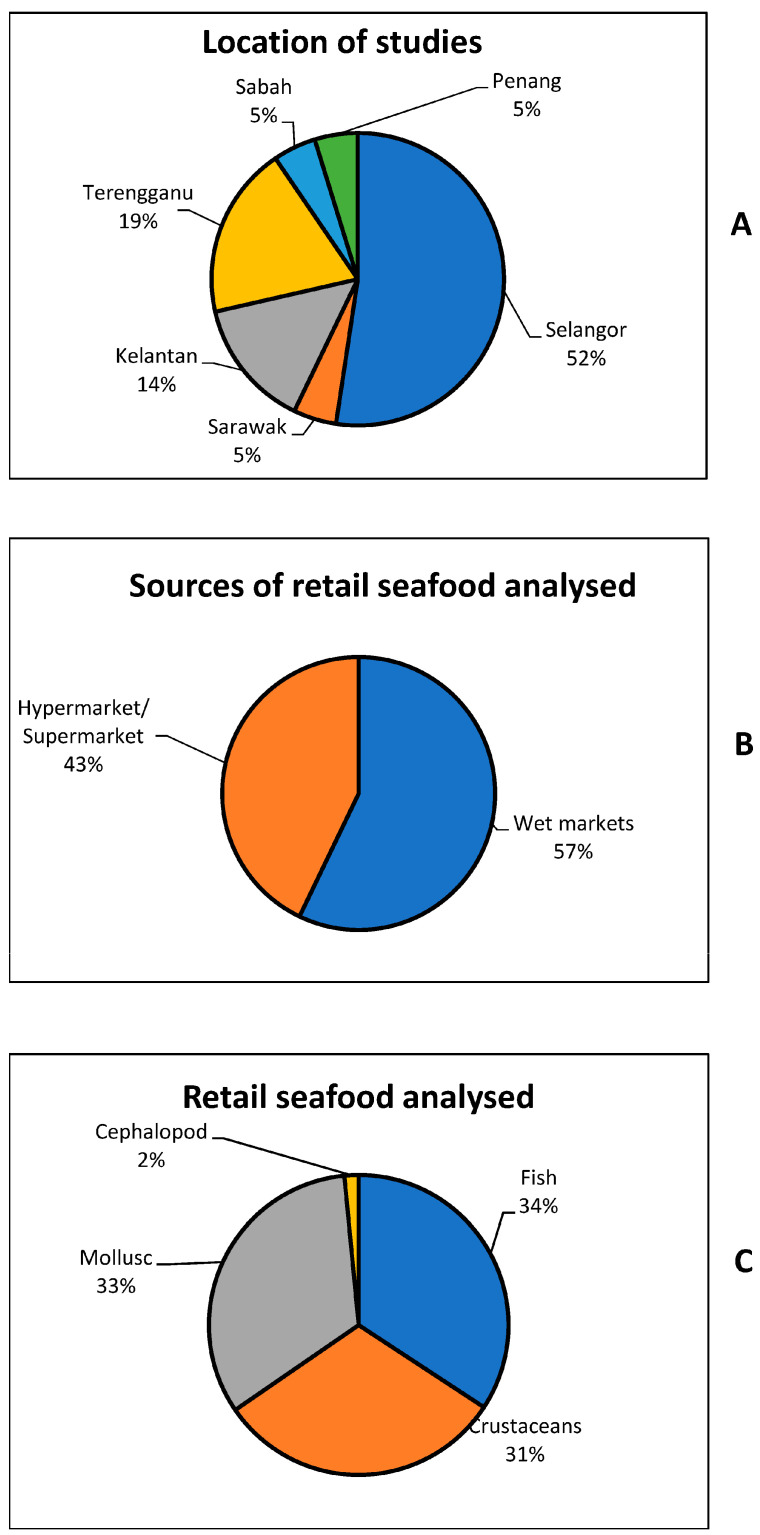
Location of studies (**A**), sources of retail seafood analyzed in the primary studies (**B**), and the retail seafood analyzed (**C**).

**Figure 3 antibiotics-12-00829-f003:**
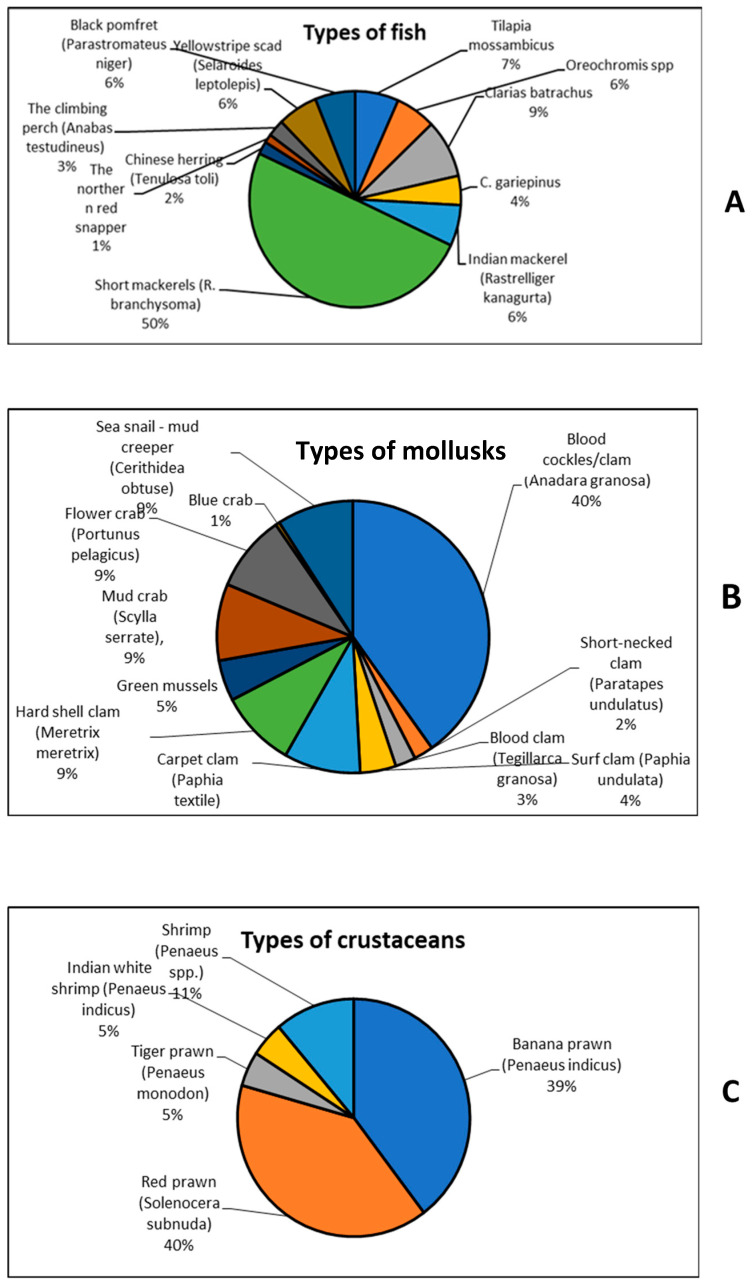
Types of (**A**) fish, (**B**) mollusks, and (**C**) crustaceans investigated for the presence of antibiotic-resistant bacteria.

**Figure 4 antibiotics-12-00829-f004:**
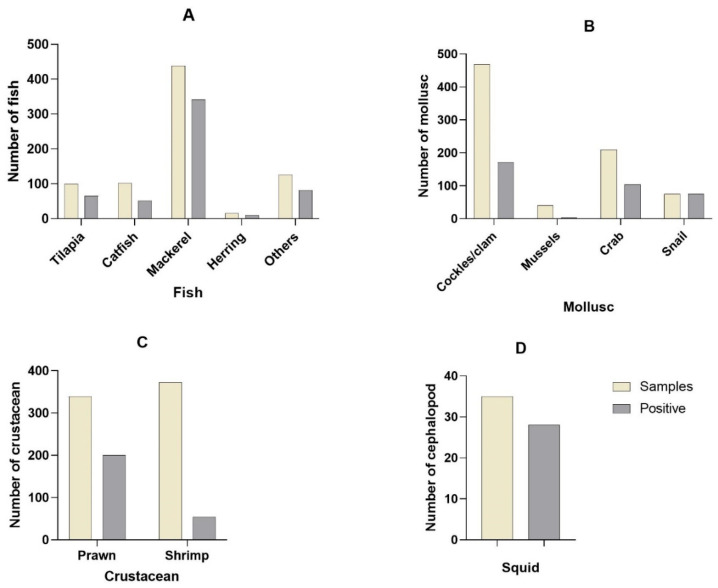
The distributions of the retail seafood were positive for pathogens in fish (**A**), mollusks (**B**), crustaceans (**C**), and cephalopods (**D**).

**Figure 5 antibiotics-12-00829-f005:**
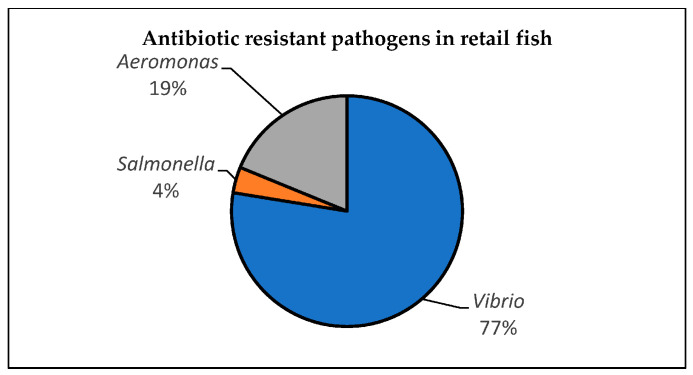
Antibiotic-resistant pathogens in retail fish.

**Figure 6 antibiotics-12-00829-f006:**
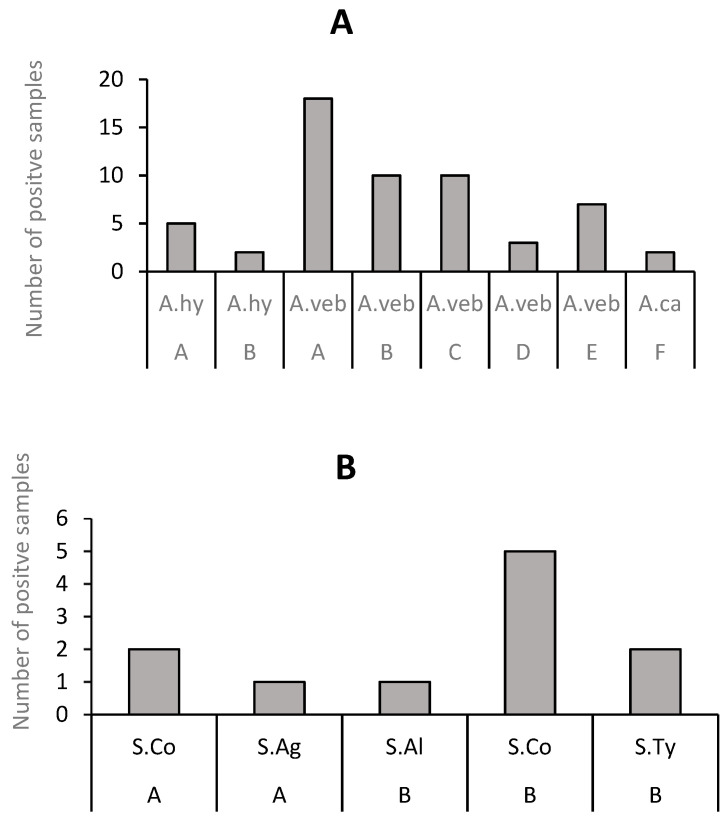
Distribution of antibiotic-resistant seafood-borne pathogens in retail fish sold in Malaysia. (**A**) Distribution of *Aeromonas* in fish: (A = *Tilapia mossambicus*; B = *Clarias batrachus*; *Tilapia mossambicus*; C = Chinese herring (*Tenulosa toli*); D = The northern red snapper; E = The climbing perch (*Anabas testudineus*); F = The northern red snapper). Pathogens: (A.hy = *A. hydrophila*; A.veb= *A. veronii* biovar sobria and A.ca = *A. caviae.* (**B**) Distribution of *Salmonella* in fish: A = *Tilapia mossambicus*; B = *C. gariepinus*; S.Ag = *S. agona*; S.Al = *S. albany*; S.Co = *S. corvallis* and S.Ty = *S. typhimurium*. (**C**) Distribution of *V. parahaemolyticus* in fish. A = *Oreochromis* spp.; B = catfish *Clarias batrachus*; C = Indian mackerel (*Rastrelliger kanagurta*); D = Short mackerels (*R. branchysoma*); E = Yellowstripe scad (*Selaroides leptolepis*) and F = Black pomfret (*Parastromateus niger*). V.pa = *V. parahaemolyticus*.

**Figure 7 antibiotics-12-00829-f007:**
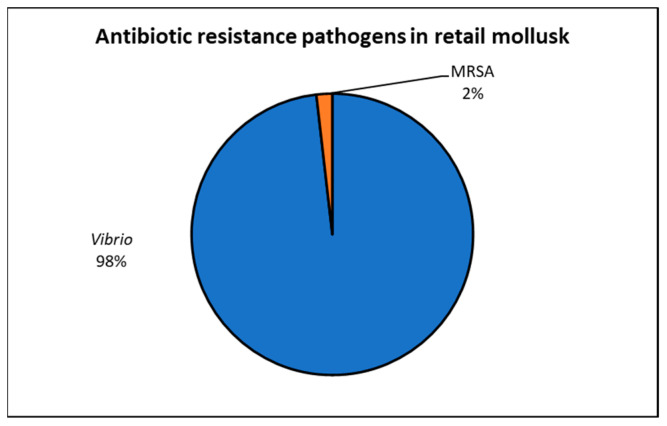
Antibiotic-resistant pathogens in retail mollusk.

**Figure 8 antibiotics-12-00829-f008:**
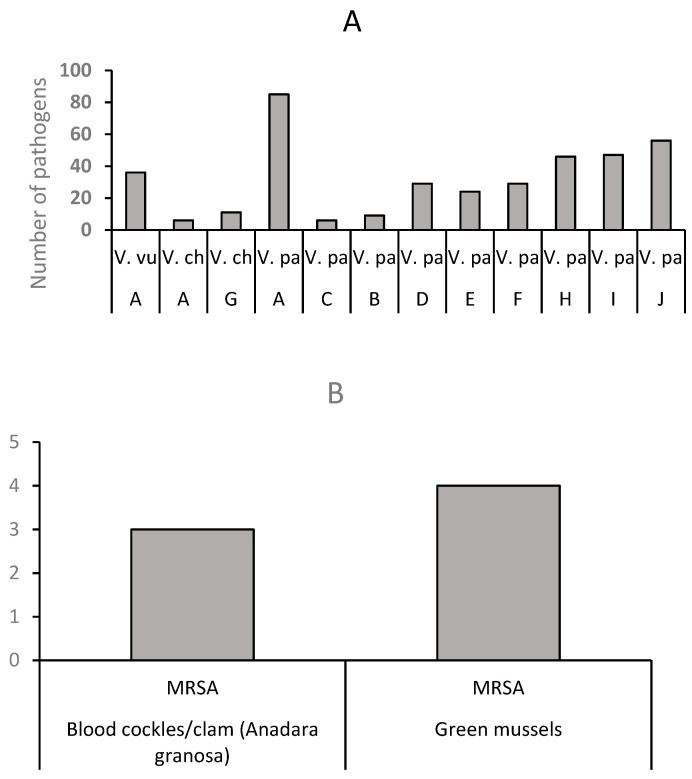
Distribution of antibiotic-resistant seafood-borne pathogens in retail mollusk sold in Malaysia. (**A**). Distribution of *Vibrio* in mollusk: A = Blood cockles/clam (*Anadara granosa*), B = Blood clam (*Tegillarca granosa*), C = Short-necked clam (*Paratapes undulatus*), D = Surf clam (*Paphia undulata*), E = Carpet clam (*Paphia textile*), F = Hard shell clam (*Meretrix meretrix*), G = Blue crab, H = Mud crab (*Scylla serrate*), I = Flower crab (*Portunus pelagicus*), J = Sea snail-mud creeper (*Cerithidea obtuse*), V.vu = *V. vulnificus*, V.ch = *V. cholera* and V.pa = *V. parahaemolyticus*. (**B**). Distribution of MRSA in mollusk: Blood cockles/clam (*Anadara granosa*), and green mussels.

**Figure 9 antibiotics-12-00829-f009:**
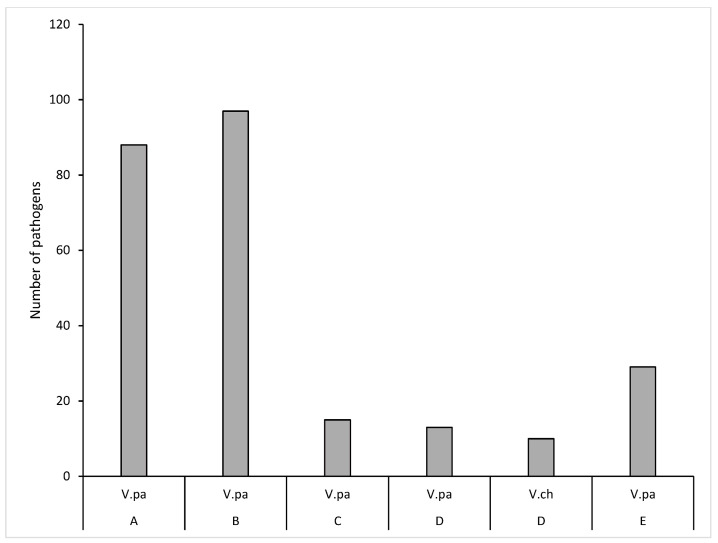
Distribution of *Vibrio* in crustaceans: A = Banana prawn (*Penaeus indicus*), B = Red prawn (*Solenocera subnuda*), C = Tiger prawn (*Penaeus monodon*), D = Indian white shrimp (*Penaeus indicus*), E = Shrimp. V.ch = *V. cholera,* and V.pa = *V. parahaemolyticus*.

**Table 1 antibiotics-12-00829-t001:** Summary of random-effect meta-analysis (NC = not completed).

Retail Seafood	Prevalence/Effect Size 95% CI	Heterogeneity			
		Q Value	df	I^2^	*p*-Value
**Prevalence of antibiotic-resistant pathogens in retail seafood in Malaysia**					
Overall prevalence	55.7% (95% CI: 0.46–0.65)	443.73	27	93.9	0
**Prevalence based on seafood types**					
Fish	62.7% (95% CI: 0.52–0.73)	61.2	10	83.7	0
Mollusks	47.5% (95% CI: 0.35–0.61)	98.3	10	89.8	0
Crustaceans	55.5% (95% CI: 0.48–0.52)	172	4	97.6	0
Cephalopods	80.0% (95% CI: 0.64–0.90)	NC	NC	NC	NC

**Table 2 antibiotics-12-00829-t002:** Types of antibiotics used across the studies (√ indicates antibiotics used in the study).

Antibiotics	[14]	[15]	[16]	[17]	[18]	[19]	[20]	[21]	[22]	[23]	[24]	[25]	[26]
Amikacin					√	√		√	√	√		√	√
Amoxicillin/clavulanic acid								√		√		√	
Ampicillin	√	√	√		√	√		√	√		√	√	√
Ampicillin-sulbactam					√	√		√	√			√	√
Azithromycin				√									
Bacitracin		√											
Carbenicillin	√	√											
Cefazolin												√	
Cefepime										√		√	
Cefoperazone		√											
Cefotaxime					√	√		√	√	√		√	
Cefoxitin												√	
Ceftaroline										√			
Ceftazidime				√	√	√	√	√	√			√	√
Ceftriaxone		√		√									
Cefuroxime											√	√	
Cephalothin		√						√				√	
Ceftazidime		√											
Chloramphenicol	√	√	√	√	√	√		√	√	√	√	√	√
Ciprofloxacin			√	√				√		√	√	√	
Clarythromycin										√			
Clindamycin				√			√						
Doxycycline								√				√	
Erythromycin	√	√									√		
Gentamicin	√	√	√		√	√		√	√	√		√	√
Imipenem					√	√		√	√			√	√
Kanamycin	√	√	√		√	√			√				√
Levofloxacin					√	√		√	√	√		√	√
Linezolid										√			
Meropenem								√				√	
Nalidixic acid	√	√	√		√	√			√				√
Norfloxacin		√	√										
Ofloxacin										√		√	
Oxytetracycline					√	√			√				√
Penicillin	√						√	√		√	√	√	
Piperacillin												√	
Piperacillin-tazobactam												√	
Quinupristin/dalfopristin										√			
Rifampicin				√			√						
Spectinomycin				√									
Streptomycin	√	√	√					√			√		
Sulfamethoxazole/trimethoprim			√	√	√	√			√			√	√
Teicoplanin											√		
Tetracycline	√	√	√	√	√	√	√	√	√		√	√	√
Tobramycin				√									
Trimethoprim							√						

**Table 3 antibiotics-12-00829-t003:** Summary of random-effect meta-analysis.

Retail Seafood	Prevalence/Effect Size 95% CI	Heterogeneity			
		Q Value	df	I^2^	*p*-Value
Prevalence of antibiotic resistance based on pathogens and type of seafood					
**Fish** (overall prevalence)	**22.2% (95% CI: 0.15–0.32)**	**943.6**	**26**	**97.2**	**0**
*Aeromonas*	51% (95% CI: 0.33–0.69)	197.7	13	93.4	0
*Salmonella*	59.9% (95% CI: 0.32–0.82)	15.21	4	67.1	0.009
*Vibrio*	10.2% (95% CI: 0.06–0.18)	774	18	97.6	0
**Mollusks** (overall prevalence)	**10.5% (95% CI: 0.07–0.16)**	**1719**	**28**	**98.4**	**0**
*Vibrio*	10.2% (95% CI: 0.07–0.15)	1661.8	28	98.3	0
MRSA	70.9% (95% CI: 0.36–0.92)	6.7	3	55.27	0.082
**Crustaceans** (overall prevalence)	**23.4% (95% CI: 0.15–0.34)**	**578.8**	**24**	**95.8**	**0**
*Vibrio*	47.5% (95% CI: 0.35–0.61)	98.3	10	89.8	0
**Cephalopods** (overall prevalence)	**67.2% (95% CI: 0.22–0.94)**	**94.1**	**12**	**87.2**	**0**
*Vibrio*	67.2% (95% CI: 0.22–0.94)	94.1	12	87.2	0

## Data Availability

Available on request.

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
