# Peer review of "Prevalence of Antibiotic-Resistant Seafood-Borne Pathogens in Retail Seafood Sold in Malaysia: A Systematic Review and Meta-Analysis"

_antibiotics, 2023, doi:10.3390/antibiotics12050829_

Round 1
Reviewer 1 Report
The above noted manuscript describes the prevalence of antibiotic resistant bacteria in seafood sold in Malaysia. There are several issues that need to be addressed in this manuscript before it can be considered for publication in Antibiotics and as such major revisions are required. Specific points are as follows:
1) the manuscript does not have flow as it is broken by the insertion of figures and tables in the text making it difficult to follow
2) there are numerous sentence structure and grammatical errors throughout the manuscript including as examples lines 41, 51, 56, 124, 202 and many more
3) Figure 1 is way too large and can be cut shrunk considerably
4) Figures 2, 3, 5, 7, appear to be powerpoint figures and the data in these figures might be better displayed in a well constructed table.
5) sadly I also feel the data in Figures 6, 8 can be better presented.
6) why does Figure 7 come before Figure 6
7) lines 148-155 need to be re-written. Specifically, when an organism name is written out, the first letter of the genus is capitalized and the first letter of the first letter of the species is lower case. Also, organism names should be in italics. The use of italics is inconsistent throughout the manuscript.
8) how can you compare antibiotic resistance concerns between studies when there was no consistency as to the antibiotics tested. this needs to be addressed in the manuscript
9) in the discussion, when using an author name in a sentence and where there is more than 2 authors, it is usually done by i.e. Ponce et al versus Ponce, Khan as you have written in line 288 and elsewhere in the discussion. If there are only 2 authors it should be Ponce and Khan...
10) did you really mean that 29% of MRSA were resistant to cefepime?
11) what do you mean by "...aquaculture practice of using mobile cephalosporinase genes..."
12) resistance is this article yet there is absolutely no information on how resistance was measured in any of the studies included in the manuscript.
I do apologize for my negative review but this manuscript requires a major rewrite including a complete restructuring of how the data is presented before it can be considered further for publication in Antibiotics. I feel the topic is important and warrants consideration for publication
Author Response
Authors: We appreciate the time and comments of the reviewer and we have provided our responses to the comments in yellow.

Reviewer 2 Report
This review well investigated the the occurrence and prevalence of antibiotic resistant seafood-borne pathogens in retail seafood sold in Malaysia. Overall, it appears to the report that simply include occurrence and prevalence data. To provide more information regarding antibiotic resistance in seafood-borne pathogens, this review should also include the antibiotic resistance profiles in relation with the use of antibiotics in aquatic culture in Malaysia and possible transfer of antibiotic resistance genes in food system from farm to table. This review also include the ways to reduce the risk of antibiotic resistance and improve seafood safety and quality.
Author Response
We appreciate the time and comments of the reviewer and we have provided our responses to the comments in yellow.

Round 2
Reviewer 1 Report
this manuscript still requires editing for grammar and sentence structure. For example, see lines 58-59 and elsewhere.
Author Response
Point 1: this manuscript still requires editing for grammar and sentence structure. For example, see lines 58-59 and elsewhere.
Response 1: The said sentence was “In Europe, antibiotic- resistant pathogens have been reported to be responsible for more than 30,000 and over 4.5 million deaths yearly [7]. It was also predicted that by 2050, antibiotic-resistant pathogens may cause 10 million deaths per year and poten-tially cause a reduction of 2% to 5% (~ 100 trillion dollars) in gross domestic products worldwide [7]”
We have now edited it to read as “In Europe, antibiotic-resistant pathogens have been reported to be responsible for more than 30,000 deaths yearly [7]. By 2050, antibiotic-resistant pathogens would have cause 10 million deaths per year globally and this could potentially cause a reduction of 2% to 5% (~ 100 trillion dollars)gross domestic products [7]” as seen in lines 56-60.
Other parts of the manuscript have also been edited as suggested.
Reviewer 2 Report
All studies should provide new information and prospecive view point. However, as responded to the reviewer's comments, as long as this is the first stage of your long project, it will be good for this level. Expected see the next stage of studies as the authors plan.
Author Response
Thanks for your comments. We look forward to the next stage of the study too.